# Experimental and Clinical Evidence Supports the Use of Urokinase Plasminogen Activation System Components as Clinically Relevant Biomarkers in Gastroesophageal Adenocarcinoma

**DOI:** 10.3390/cancers13164097

**Published:** 2021-08-14

**Authors:** Gary Tincknell, Ann-Katrin Piper, Morteza Aghmesheh, Therese Becker, Kara Lea Vine, Daniel Brungs, Marie Ranson

**Affiliations:** 1Illawarra Health and Medical Research Institute, Wollongong, NSW 2522, Australia; gwt714@uowmail.edu.au (G.T.); akpiper@uow.edu.au (A.-K.P.); morteza@uow.edu.au (M.A.); kara@uow.edu.au (K.L.V.); Daniel.Brungs@health.nsw.gov.au (D.B.); 2Illawarra Cancer Care Centre, Illawarra Shoalhaven Local Health District, Wollongong, NSW 2500, Australia; 3School of Chemistry and Molecular Biosciences, University of Wollongong, Wollongong, NSW 2522, Australia; 4School of Medicine, University of Wollongong, Wollongong, NSW 2522, Australia; 5Ingham Institute for Applied Medical Research, Liverpool, NSW 2170, Australia; Therese.Becker@inghaminstitute.org.au; 6UNSW Medicine, University of New South Wales, Kensington, NSW 2052, Australia; 7School of Medicine, Western Sydney University, Sydney, NSW 2560, Australia

**Keywords:** urokinase plasminogen activator (uPA), urokinase plasminogen activator receptor (uPAR), plasminogen activator inhibitor type 1 (PAI-1), circulating tumour cell (CTC), biomarkers, gastric cancer, oesophageal cancer, serine proteases, tumour microenvironment, serpins

## Abstract

**Simple Summary:**

Patients with gastric and oesophageal adenocarcinomas (GOCs) have short life expectancies as their tumours spread to other sites early. This is facilitated by the increased expression of the urokinase plasminogen activation system (uPAS); a feature of the majority of GOCs. There is increasing appreciation of the importance of uPAS expression in a range of cell types within the tumour microenvironment. Abundant clinical evidence indicates that altered expression of uPAS proteins is associated with worse outcomes, including time to tumour recurrence and patient survival. Emerging technologies, including liquid biopsy, suggest a role of uPAS for the detection of circulating tumour cells, which are responsible for the dissemination of cancers. We review and summarise pre-clinical and clinical data that supports the use of uPAS as a biomarker in GOC.

**Abstract:**

Gastric and oesophageal cancers (GOCs) are lethal cancers which metastasise early and recur frequently, even after definitive surgery. The urokinase plasminogen activator system (uPAS) is strongly implicated in the invasion and metastasis of many aggressive tumours including GOCs. Urokinase plasminogen activator (uPA) interaction with its receptor, urokinase plasminogen activator receptor (uPAR), leads to proteolytic activation of plasminogen to plasmin, a broad-spectrum protease which enables tumour cell invasion and dissemination to distant sites. uPA, uPAR and the plasminogen activator inhibitor type 1 (PAI-1) are overexpressed in some GOCs. Accumulating evidence points to a causal role of activated receptor tyrosine kinase pathways enhancing uPAS expression in GOCs. Expression of these components are associated with poorer clinicopathological features and patient survival. Stromal cells, including tumour-associated macrophages and myofibroblasts, also express the key uPAS proteins, supporting the argument of stromal involvement in GOC progression and adverse effect on patient survival. uPAS proteins can be detected on circulating leucocytes, circulating tumour cells and within the serum; all have the potential to be developed into circulating biomarkers of GOC. Herein, we review the experimental and clinical evidence supporting uPAS expression as clinical biomarker in GOC, with the goal of developing targeted therapeutics against the uPAS.

## 1. Introduction

Gastroesophageal cancers (GOC) are amongst the leading causes of cancer related morbidity and mortality worldwide [1]. Gastric cancers are ranked fifth for incidence and third for deaths worldwide [1]. Oesophageal carcinomas join gastric cancers in the global top 10 for both incidence (9th) and mortality (6th) [1]. GOCs often present at an advanced stage owing to its aggressiveness and early metastasis formation, with 25–50% of GOC presenting as metastatic at diagnosis [2,3,4]. Henceforth, *GOC* will refer to adenocarcinomas arising from any location within the oesophagus or stomach, otherwise individual locations will be identified.

The plasminogen activation system is a multi-component regulatory system that, under normal conditions, functions in the clearance of blood clots and degradation of the extracellular matrix (ECM) and basement membranes (BM) during tissue remodelling processes such as wound healing [5,6,7,8,9,10]. However, unregulated plasminogen activation via the urokinase plasminogen activator (uPA) is implicated in key events in tumour progression, specifically solid tumour invasion and metastasis [5,6,7,8,9,10]. Through binding of uPA to the uPA receptor (uPAR), which is typically cell-surface-bound, co-localised plasminogen is converted to plasmin [6,11]. As a broad-spectrum serine protease, plasmin then directly and indirectly (via the activation of pro-metalloproteinases) degrades a wide range of proteins in the ECM and BMs. This process enables tumour cell dissemination around the body, a key step required for the seeding of tumour cells at distant sites to form metastases [6,7,11,12,13,14,15]. Tissue plasminogen activator (tPA) is the intra-vascular counterpart to uPA, involved in fibrin degradation to prevent blood clot formation [13]. However, tPA does not appear to play a significant role in the development of solid tumours [16]. 

Overexpression of components of the uPA system (uPAS) in GOCs, on tumour cells and/or associated stromal cells in the tumour microenvironment (TME), is strongly associated with worse tumour staging [17,18,19,20], clinicopathological features [21,22,23,24,25,26,27] and reduced patient survival [12,17,18]. Here we review the important role the uPAS plays in the development and progression of GOC and summarise the available evidence of its role as a biomarker in GOC. 

## 2. Major Components and Function of the uPAS

The key components of the uPAS include uPA, uPAR, plasminogen and specific uPA and plasmin inhibitors. uPA is a single stranded extracellular protein, secreted as an inactive double stranded zymogen (pro-urokinase), which is produced by leucocytes, fibroblasts and the urogenital system in normal physiological conditions [7]. Upon binding to its receptor uPAR, pro-uPA is converted to active uPA by proteolytic cleavage via plasmin and potentially cathepsin, plasma kallikrein or mast cell tryptase in the TME, resulting in the conversion of co-localised plasminogen to plasmin (via a number of potential cell-surface localised proteins containing c-terminal lysins) (reviewed by Ranson and Andronicos [6]). This positive feedback loop of plasmin-mediated pro-uPA activation and uPA-mediated plasminogen activation, results in increased proteolytic activity at the cell surface which is protected from inhibition by plasmin-specific inhibitors (e.g., α2-antiplasmin) [6,13,28]. Bound plasmin then also cleaves a range of multiple downstream extracellular targets, including ECM proteins such as fibrin, fibronectin and laminin and pro-metalloproteinases (pro-MMPs) (reviewed by Deryugina and Quigley [29]). Plasmin and MMP activity can also regulate cellular growth and migration through cleavage of extracellular components to release or activate chemokines, cytokines and growth factors (e.g., hepatocyte growth factor (HGF)/scatter factor, macrophage-stimulating protein, transforming growth factor (TGF) and basic fibroblast growth factor) [30,31] (Figure 1).

uPAR is a heavily glycosylated protein and is either membrane-bound via a glycosyl-phosphatidylinositol anchor or found in its soluble forms [7]. uPAR consists of three similarly structured domains, made up of approximately 90 residues each, with domain 1 (D1) responsible for uPA binding leading to plasmin generation at the cell surface. Through complex direct and indirect interactions with a range of binding partners (including vitronectin, integrins, growth factor receptors and others), uPA-bound uPAR can also modulate downstream cell signalling pathways (Figure 1) [31,32,33]. Thus, the combined proteolytic and signalling outputs of the uPAS activate many downstream events driving ECM degradation, cell proliferation, adhesion and migration. 

Soluble uPAR (suPAR) is produced through cleavage of the membrane bound uPAR; this cleavage occurs between the glycosylphosphatidylinositol (GPI)-anchor molecule and domain 3 of uPAR facilitated by plasmin, cathepsin G and GPI-specific phospholipase-D and can be identified in plasma, ascites and urine [34]. Vascular endothelial cells, monocytes and neutrophils are all known producers of suPAR [35]. Three detectable subgroups of suPAR have been identified: intact suPAR (I-III), domain 1 (D1)-suPAR(I) and intact and cleaved domains 2 (D2) and 3 (D3)-suPAR (I-III)+(II-III) [36]. suPAR (I-III) competes with membrane-bound uPAR for binding to uPA through its D2 and D3 domains and maintains its cell adhesion role through vitronectin binding with the D1 domain [37,38]. Fragmented suPAR (suPAR (I) and suPAR (II-III)) lose their ability to bind with vitronectin, resulting in reduced cell adhesion [38]. D1 is required for uPA binding, however suPAR (I) alone has low affinity for uPAR in the absence of D2 and D3 [39]. suPAR (II-III) has been shown to be a chemotactic molecule through 7TM receptor FPR-like receptor 1, attracting immune cells to cancers [40,41,42].

A key level of control in the regulation of plasmin activity arises through inhibition of uPA (and tPA) via the serine proteinase inhibitors (serpins) plasminogen activator inhibitor (PAI)-1 and PAI-2 (Figure 1). While the expression of both PAI proteins can be stimulated by various factors, including inflammatory conditions, under normal physiological conditions PAI-1 is mainly produced by endothelial cells and PAI-2 by synciotrophoblasts of the placenta in late pregnancy [13]. Activation of uPAS, such as infection and inflammation, results in increased PAI-1 expression in fibroblasts, adipocytes, smooth muscle cells and macrophage cells, whereas increased PAI-2 expression is detected in endothelial cells, macrophages, monocytes and platelets [11,43]. Both PAI-1 and PAI-2 irreversibly bind to and inhibit uPAR-bound uPA [11]. The uPA-PAI/uPAR complex is then taken up into the cell via low density lipoprotein receptor-related protein-mediated endocytosis [44,45,46]. uPAR is then recycled to the cell surface for further uPA interaction [44,45,46]. The two PAI proteins bestow different effects on cancer: cancers with high PAI-1 expression have been consistently demonstrated to have poorer clinical outcomes, whereas the effect of elevated PAI-2 expression levels are less well defined and the impact on clinical outcomes less pronounced [11,47]. Even though both PAI proteins mediate uPA/uPAR endocytosis, there are clear differences in functional outcomes from these interactions with endocytosis receptors [46,48]. For example, PAI-2 inhibits and clears cell surface uPA (and hence proteolytic activity) without influencing the promitogenic signalling pathways activated via PAI-1 [48]; this has been explained by distinct structural elements that underlie the interactions of these serpins with endocytic receptors [46]. PAI-1 also has established roles in various other cancer-promoting activities including resisting tumour cell death, increased cell migration and angiogenesis, via a variety of mechanisms that affect cell adhesion and signalling pathways (reviewed in detail by Kubala and Declerck 2019 [47]). Thus, while both serpins have anti-plasminogen activation activity, and loss or gain of PAI-2 expression has been shown in a cancer context-specific manner to be associated with worse or improved outcomes, respectively [11]; the clinical data showing that increased PAI-1 expression is strongly correlated with poor cancer outcome is highly convincing [47]. Moreover, PAI-1 levels can predict a response to chemotherapy in breast cancer, with increased PAI-1 levels associated with improved outcomes following administration of chemotherapy [49]. PAI-1 is thus also considered an important cancer biomarker.

## 3. Regulation of the uPAS

The expression and activity of the uPAS is tightly regulated during physiological processes to prevent unnecessary ECM remodelling through the production of excessive plasmin at the cell surface and dysregulated downstream signalling [11,50,51]. Certain cells secrete uPA and express uPAR at low levels [52] however, hormones [53,54,55,56], growth factors [55,57,58,59], cytokines [60,61] and tumour promoters [62,63,64], which also affect cellular proliferation and differentiation, induce overexpression of these components [65] in a variety of cancer cell lines.

Key cancer signalling pathways also alter uPAS mRNA and protein expression in GOC cell lines and xenografts. uPAS expression can be modulated by targeting key pathways with drug blockade [66,67], transfection of interfering or promoting RNA [68,69,70,71] and exposure to exogenous stimulating proteins [72,73,74,75,76,77]. Table 1 summarises the molecules and pathways linked to the regulation of uPAS in GOC cell lines.

Exposure to exogenous growth factors such as epidermal growth factor increases uPAR mRNA expression, and this appears to occur through the mitogen-activated protein kinase (MAPK)/extracellular signal-related kinases (ERK) signalling pathway [72]. uPA and uPAR are both also upregulated upon HGF exposure, again, uPAS expression is reportedly linked to the MAPK/ERK pathway rather than phosphoinositide 3-kinase (PI3K) pathway [70]. Increased prostaglandin E2 levels (including as a result of nicotine exposure) resulted in increased uPA and uPAR levels via the cyclooxygenase-prostaglandin pathway [74,75]. Upregulation of transforming growth factor-beta (TGF-β) pathway also results in increased uPA and uPAR expression via the MAPK/ERK but also via the PI3K and Jun-N-terminal kinases pathways [67,71,76].

## 4. The Clinical Relevance of uPAS Expression in GOCs

The expression of uPAS in tumour tissue, stroma and liquid biopsies correlates to both clinicopathological features of tumours [18,19,20] and patient survival data [12,17,18,27]. In general, the assessment of the uPAS relies on protein or mRNA expression and levels, opposed to assessment of the function (or activity) of the individual proteins of the system. Immunohistochemistry (IHC) and enzyme-linked immunosorbent assay (ELISA) are the most used methods for protein assessment. 

### 4.1. Tumour Expression and Association with Clinicopathological Features

A meta-analysis by Brungs et al. evaluated uPAS expression in GOCs, which demonstrated the following expression levels: uPA 52.8%, uPAR 56.8%, PAI-1 53.3% and PAI-2 57.5% of all patients with GOC [17]. Reporting of PAI-2 expression in oesophageal adenocarcinomas (via ELISA) is variable with some studies showing reduced levels and downregulation [81,82]. 

Activation of the uPAS system is a requirement for tumour cells to invade deeper into the ECM or seed at distant metastatic sites [83]. Therefore, it is not unexpected to find that increased expression of the uPAS proteins in GOC is associated with worse clinicopathological features including depth of invasion (T score), presence of metastasis (N score-lymph nodes, M score-distant metastasis) and histological grade of disease (Table 2). 

Relative uPA expression levels are biologically important: where >20% of primary tumour cells stained positive for uPA, higher tumour staging and histological grading was seen [84]. As will be discussed below, the combination of uPA and PAI-1 has been shown to be useful as biomarkers of worse prognosis, however one study found that PAI-1 negative, highly uPA-expressing gastric adenocarcinomas were associated with increased volume and number of metastases [22]. Comparison of high nodal and low nodal stage III gastric adenocarcinomas confirmed *SERPINE1* gene expression (encoding for PAI-1) was higher in those patients with increased nodal disease (>2x compared to healthy tissue) [85]. It can thus be concluded that upregulated PAI-1 expression is an important regulator of malignant lymph node development [85].

To date, PAI-2 alone has not been associated with any clinicopathological features as described in Table 2 [22,86]. However, advanced clinical staging of GOCs is associated with high uPA protein expression but absence of PAI-2 [86]. Gastric adenocarcinoma patients with a higher nodal status (>5 involved lymph nodes) was seen with low PAI-2 protein expression [87]. A lack of PAI-2 is therefore likely to be associated with worse tumour staging in combination with other uPAS protein dysregulation. 

The peritoneum of patients with GOC peritoneal metastases shows generalised uPAS upregulation compared to uninvolved peritoneum of patients with GOC metastases at other sites [88]. uPAS expression (uPA, uPAR, PAI-1), however, did not alter between malignant and non-malignant peritoneum within the patients with peritoneal metastases [88]. Translational investigations confirm the role of altered uPAS expressing cell lines in the development of peritoneal metastasis [89] and increased ascites formation [90].

Retrospective analysis of GOCs with lymph node metastases showed uPAS protein expression in the primary tumour was correlated with lymph nodal metastases [19]. 82% of patients with malignant lymph nodes had strong uPA expression in the primary gastric cancer (IHC ≥ 50%), while in lymph node-negative disease, the primary cancer only showed uPA expression in 52% of cases [19]. uPAS expression in malignant lymph nodes demonstrates the critical role of uPAS in tumour invasion at secondary sites [18,20]. 

### 4.2. Tumour Expression and Association with Clinical Outcomes

uPAS overexpression is associated with poorer disease-free and overall survival (OS) of patients with GOCs (meta-analysis results of IHC, ISH and ELISA shown in Table 3) [17]. In individual studies, uPAS expression showed variable strength of association with prognosis (reviewed by Brungs et al., 2017 [17]). 

Subgroup analysis of uPAS expression in intestinal and diffuse gastric adenocarcinomas was assessed by Heiss et al. [91]. In this study uPA and uPAR were assessed on intestinal-type gastric adenocarcinomas and could not be associated with prognosis or recurrence-free survival; however, PAI-1 overexpression was an independent factor for recurrence-free survival [91]. In diffuse-type gastric adenocarcinoma, overexpression of uPA, uPAR and PAI-1 was associated with poorer overall- and recurrence-free survivals [91]; PAI-2 showed association with OS but not recurrence-free survival [91]. These subgroup findings may not be truly representative due to possible under powering with reduced numbers in the subgroup analysis.

Oesophageal adenocarcinomas that show uPAS overexpression are associated with poorer prognosis with elevated uPA protein levels shown to be associated with reduced median OS [24].

### 4.3. Intra-Tumoural Heterogeneity

uPAS expression shows significant intra-tumoural heterogeneity in GOC and can vary widely within patients, within the same tumour, between the primary tumour and its metastatic tumour or between different tumour histology types. For example, Alpízar-Alpízar et al. demonstrated uPAR overexpression at the invading front of gastric adenocarcinomas but not the tumour core [12]. This expression pattern was significantly associated with poorer OS in multivariate analysis (Hazard ratio (HR) = 2.39; 95% confidence interval (CI): 1.22–4.69; *p* = 0.011) [12].

We have investigated uPAR expression at the tumour core and invasion front in an Australian cohort of GOC patients [92]. uPAR IHC was assessed by an experienced anatomical pathologist with the following cut off values: 0—no uPAR positive cells, 1—less than 1% uPAR positive cells, 2—1–5% uPAR positive cells, 3—5–10% uPAR positive cells and 4—more than 10% uPAR positive cells. We found that increased uPAR expression at the invasion front (uPAR IHC 0–1 vs. >2) was significantly associated with worse patient survival (Figure 2a). uPAR expression within the tumour core was not significantly associated with OS (Figure 2b).

uPAR overexpression at the invading front of tumours has been supported in a number of other studies [18,21]. Increased uPA [22] and PAI-1 [21] expression at the leading edge of the cancer is also recognised. The higher uPAS expression at the invasion front is critical to facilitate tumour progression through the surrounding stroma. 

### 4.4. Expression in Tumour-Associated Stromal Cells

The invasion of cancer cells into normal tissues relies on interactions between the tumour and the surrounding stroma. There is increasing evidence of the importance of stroma in initiating and regulating the speed of invasion [93,94,95]. The stromal cells within the TME of particular interest are immune cells such as leucocytes and tumour-associated macrophages, as well as fibroblasts, blood- and lymphatic endothelial cells [96]. uPAS overexpression is seen in the immediate adjacent stromal cells where it assists in the degradation of the stromal laminin and fibronectin [12]. 

As expected, the most critical tumour region for uPAS expression in the stroma is at the advancing tumour front. Macrophages and myofibroblasts at the invading front of GOCs express increased uPAR compared to the tumour core [12,18]. In adenocarcinomas arising from the oesophagus, gastroesophageal junction and cardia, strongly uPAR-expressing macrophages at the invasion front are inversely correlated to OS when compared to those with lower expressing macrophages (multivariate, HR 6.26, 95%CI 2.37–16; *p* = 0.0002) [18]. This was not replicated in distal gastric adenocarcinomas [12]. Conflicting results may be due to the dual role of macrophages in tumours as either pro-tumourigenic or anti-tumourigenic; thus, assessment of uPAR alone may be insufficient to describe the role of macrophages in cancer progression [97]. In addition, intra-observer variability in assessing uPAR expression was high which may have confounded results [18]. 

uPAR-expressing myofibroblasts (defined by expression of α-smooth actin) are not significantly associated with patient outcomes in GOC [12,18]. However, further work is needed to clarify if uPAR expression on the population of so-called cancer-associated fibroblasts, a fibroblast subpopulation which are more likely to be involved in cancer modification, is prognostic.

Similarly, in other solid tumours, stromal uPAS expression is significantly linked with tumour-associated stromal cells, and in the case of colon cancer poorer clinical outcomes [27]. There is evidence in breast-, colon- and lung cancer of strong association of uPAS expression on both macrophages and fibroblasts (Table 4). In colon cancer, there is further supporting evidence of stromal uPAS expression being inversely associated with disease free survival times (multivariate HR 1.71, 95%CI 1.05–2.80; *p* = 0.002), and a tendency to worse OS (*p* = 0.07) [27].

### 4.5. Interactions of the uPAS with Other Proteolytic Enzymes 

There are many MMPs with different functional roles, with variable association with cancer occurrence and progression [106]. In one study of gastric adenocarcinoma, both uPA and MMP-9 mRNAs were shown to be expressed in 58% of tumours, but co-expression was not explored [107]. However, both uPA and MMP-9 were shown to be independent prognostic factors, in addition to standard prognostic tumour features [107]. Co-expression of MMP-2 with uPA, uPAR, PAI-1 or PAI-2 is seen in gastric cancer, with co-expression of MMP-2 and uPAR associated with worse OS [108]. Gastric adenocarcinoma tissues overexpressing MMP-2 mRNA are associated with lymph node metastases, histological differentiation and diffused or mixed Lauren’s classification when compared to normal adjacent tissues [109]. 

Cathepsin B is a cysteine protease which has indirect proteolytic activity through interactions with pro-uPA, pro-MMPs, TGF-β and toll-like receptor 3, therefore it has an important role in cell proliferation, differentiation and angiogenesis [110,111]. Cathepsin B is localised in mitochondria and here it initiates apoptosis [111]. Cathepsin B helps catalyse pro-uPA to its active form urokinase [112]. Serum Cathepsin B and soluble uPA levels were shown to be higher in gastric cancer patients when compared to patients with premalignant adenomas, which were higher again than normal controls [113]. Increased serum levels of both Cathepsin B and uPA were also seen in metastatic compared to localised GOCs [113]. 

## 5. uPAS Assessment in Blood 

Peripheral blood sampling allows for minimally invasive assessment of patient’s tumour and immune response. The uPAS has been assessed in serum [35,36,114,115,116,117], immune cells [118] and circulating tumour cells (CTCs) [92]. However, assessment of peripheral circulating uPAS proteins in serum or plasma can be complicated by elevated uPAS expression levels seen in non-malignant conditions including renal failure, sepsis, inflammatory arthritis and cardiovascular disease [119,120]. Overall, there is poor correlation of each individual uPAS protein assessed in the plasma and primary cancer tissue samples in patients with gastric adenocarcinoma, with plasma uPAS levels not associated with cancer staging or severity [121]. However, higher uPAR mRNA levels were seen in the peripheral blood of patients with gastric cancer compared to those with benign gastric diseases and the mRNA levels were also associated with more advanced tumour stages [114].

### 5.1. Soluble uPAS Proteins in the Serum

To date, only two studies have investigated the role of serum uPA levels in GOC with inconsistent findings. Herszényi, et al. showed serum uPA levels were associated with a diagnosis of GOC and the severity of disease [113]. However, Vidal, et al. showed serum uPA levels in surgically curative gastric adenocarcinoma patients compared to healthy controls were comparable, with no significant associations seen with pathological features or clinical outcomes [115]. The lack of serum uPA discrimination may be due to the early stage of these cancers or participant selection. The prognostic role of blood uPA levels were however reported in advanced and metastatic breast cancer [122].

ELISA [35] and time-resolved fluoroimmunoassay [123] are both methods which are available for detection of suPAR. However, neither of these techniques are used routinely in clinical practice and would currently be considered for research use only. In GOC patients, levels of all suPAR subunits were reported at almost double that of aged-matched healthy individuals ([ng/mL] 5.74 ± 5.3 vs. 2.77 ± 0.77; *p* < 0.0001), and significantly higher in those with metastatic disease compared to non-metastatic disease ([ng/mL] 7.00 ± 6.13 vs. 4.75 ± 4.43; *p* > 0.05) [35,36]. In vitro models have shown that tumour-associated suPAR can direct migration, promote mitosis and angiogenesis of human umbilical vein endothelial cells demonstrating the potential role of suPAR in the progression of tumours [116].

suPAR has been better characterised in other gastrointestinal cancers. In colon cancer, increased pre-operative suPAR levels are significantly associated with poorer prognosis [117]. Interestingly, the dynamics of suPAR also appear important. In patients with paired pre-operative and six-month post-operative suPAR recordings, a rising suPAR level was associated with shorter survivals, while the converse was seen for those with a falling post-operative suPAR [124]. Those patients with highest suPAR levels following liver metastases had worse prognosis [124]. Increased levels of suPAR are postulated to be a product of more aggressive cancer and demonstrating non-radiological invasive disease, hence it has potential as a prognostic biomarker.

### 5.2. uPAS Expression on Peripheral Blood Mononuclear Cells in GOC

Peripheral blood mononuclear cells (PBMCs), identified by gradient centrifugation of blood, includes the majority of leukocytes. In malignancy, monocytes may display both pro-tumoural and anti-tumoural effects on cancers [118]. As such, assessment may be able to aid prognostic decision making.

uPA mRNA assessed in peripheral blood monocytes in treatment naïve patients, following gastrectomy, demonstrated that patients with more advanced disease showed higher relative levels prior to adjuvant chemotherapy (stage III vs. I or II; *p* = 0.014) [125]. OS was also significantly reduced in patients with uPA mRNA expression above the median value (*p* = 0.014) [125].

### 5.3. Evidence of uPAS on Circulating Tumour Cells

In addition to leukocytes, the PBMC layer also contains CTCs, which are a critical link in the development of distant metastases. High CTCs numbers in GOCs show worse prognosis and poor response to therapy [126,127].

Current food and drug administration agency approved CTC isolation utilises epithelial cell adhesion molecule (EpCAM) expression as a positive marker for CTCs [128]. EpCAM is a marker of the epithelial phenotype and, as such, may not capture CTCs that have undergone epithelial to mesenchymal transition (EMT) [128]. uPAR is a known translocator of cells to the mesenchymal phenotype [129]. Given the propensity of cells at the invasive front in GOC to overexpress uPAR (Figure 2) and likely give rise to CTCs that have undergone EMT, uPAR has the potential to be utilised as an alternate CTC capture target molecule.

Brungs et al. assessed 43 patients from whom CTCs were isolated using the standard EpCAM isolation methods at any clinical stage of GOC. In 93% of patients, where EpCAM selected CTCs were identified, a proportion also co-expressed surface uPAR (CK+/EpCAM+/DAPI+/CD45-/uPAR+ CTCs) [130]. In further analyses, we found that where more than 60% of these EpCAM selected uPAR+ CTCs also co-expressed uPAR, histological tumour uPAR IHC scoring was also increased (Figure 3). Metastasis formation and OS was not associated with proportional assessment of CTCs (more than 60% of EpCAM-selected showed uPAR-positivity) in this cohort of patients (Figure 4b). There was a trend to poorer OS in this highly selected group of patients where absolute number of EpCAM+/uPAR+ CTCs was greater than 10, but this would likely be attributed to absolute higher CTC numbers opposed to proportional cut offs (Figure 4a). Intriguingly, higher CTC numbers may be linked to uPAR-positivity; however, any such connection needs to be more closely investigated.

This data shows the feasibility of uPAR detection on CTCs captured targeting cell surface EpCAM. Given the proposed role of uPAR in promoting aggressive tumour phenotypes, further investigation of uPAR in CTC development and their ability to form metastasis is certainly warranted.

## 6. Therapeutics and Diagnostics Directed towards the uPAS Pathway

It is clear that in many carcinoma types, including GOC, the uPAS is a driver of tumour aggressiveness. Not surprisingly, several experimental anti-cancer and imaging approaches targeting various components of the uPAS have been pursued (reviewed in detail by Lin et al. [131], Mahmood and Rabbani [15] and Yuan et al. [132]). Briefly, anti-uPAS therapeutic approaches include antagonists of uPAR and various uPAR ligand (e.g., uPA, vitronectin, integrins, etc) interactions, small molecule and antibody inhibitors directed against uPA protease activity, PAI-1 inhibitors and uPA-therapeutic drug conjugates [15,131,132]. Our group has previously described the use of PAI-2 conjugated cytotoxins and therapeutic radioisotopes, which were effective in mouse models of human breast and colon cancer [133,134,135,136,137]. We have also recently described novel amiloride analogues with low nanomolar uPA inhibitory activity, high target selectivity and potent antimetastatic activity in mouse models of human lung and orthotopic pancreatic cancer metastasis [138,139]. To date, most of these experimental approaches have not progressed beyond pre-clinical models and very few have utilised models of GOC [68,140,141]. One orally active small molecule uPA inhibitor upamostat (the prodrug form of WX-UK1) was efficacious in a Phase 2 trial for locally advanced non-resectable pancreatic cancer in combination with gemcitabine showing a 17% increase in 1-year survival over gemcitabine or upamostat alone and an acceptable safety profile [142]. However, upamostat shows broad activity across many serine proteases and is currently being tested in other indications including a Phase 2/3 study for patients with symptomatic COVID-19 (NCT04723537). Nevertheless, this highlights the promise of perhaps more selective uPAS drugs for the treatment of advanced disease. Small molecule uPAR binding peptides and antibodies targeting uPAR and uPA conjugated to imaging radioisotopes are also being developed that have been shown to successfully detect primary tumours and metastases (which overexpress uPA/uPAR) with ongoing clinical trials aiming to determine the utility of these approaches for prognostication and/or response to therapy (reviewed in Mahmood and Rabbani [15]). To the best of our knowledge, none of these trials yet includes patients with GOC.

## 7. Conclusions

The uPAS is an important pathway whose upregulation contributes to uncontrolled ECM remodelling and cell signalling resulting in increased tumour, invasion and metastasis. uPA, uPAR and PAI-1 all have clear prognostic associations with GOCs, with evidence supported by a multitude of individual studies and a meta-analysis. Further, the expression of uPAS is associated with adverse clinicopathological features of GOCs. Therefore, GOC tumour levels of uPA, uPAR and PAI-1 can be considered a significant prognostic biomarker, with increased expression resulting in worse outcomes for patients.

Tumour-associated stroma is infiltrated with immune cells; the role of this stroma is the focus of ongoing research uncovering a deeper understanding of its role in tumour progression. uPAS expression is elevated in macrophages and myofibroblasts in GOC. GOC (except distal gastric) stromal macrophage uPAR expression is associated with a poorer prognosis. The role of the uPAS in the stroma is under-investigated in GOCs; larger cohorts and prognostic assessment are required to understand the role of the uPAS protein expression in the stroma.

An optimal biomarker for GOCs would offer real time prognostic and/or predictive qualities, as such liquid biopsy is of keen interest. suPAR shows promise as a diagnostic biomarker with increased expression reported in patients with GOCs. Unfortunately, to date, suPAR has failed to yield a prognostic association in GOCs in the same way as it has for colon cancer. For this reason, it is not currently considered a useful predictive biomarker. uPAR mRNA isolated from circulating immune cells from the peripheral blood monocyte layer has been shown to have prognostic potential, when assessing reduction in OS. These findings offer potential of a uPAS-related prognostic biomarker being identified in the circulating blood.

In summary, uPAS has a highly active role in the progression of GOC, and compelling evidence of its relationship with prognosis and clinicopathological features regardless of its assessment in the primary tissue or as a circulating biomarker. GOC uPAS expression in tumour-associated stroma requires further investigation to further specify the stromal role in tumour progression. We have demonstrated primary tissue assessment of the uPAS as a useful prognostic biomarker in GOCs and highlighted the exciting potential of liquid biopsies to be added to the list of prognostic biomarkers. Through ongoing investigation and drug development to target this pathway, there is significant potential for the uPAS as a predictive biomarker of uPAS directed therapies.

## Figures and Tables

**Figure 1 cancers-13-04097-f001:**
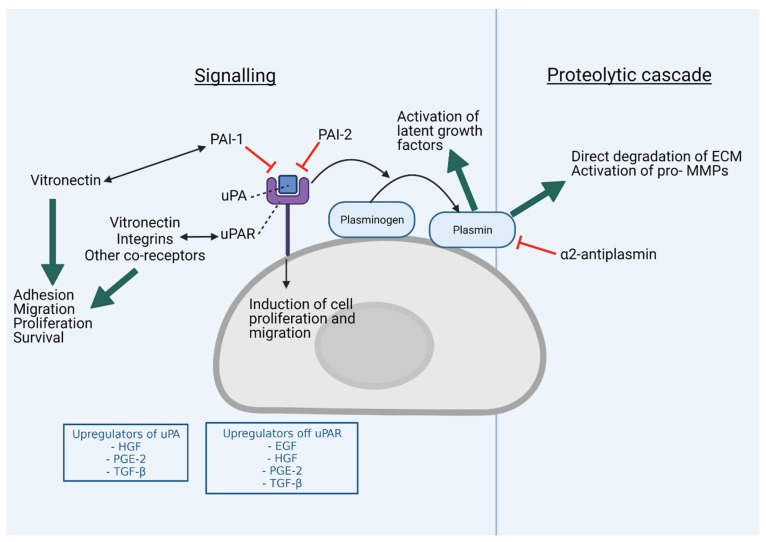
Overview of the urokinase plasminogen activation system. The binding of urokinase plasminogen activator (uPA) to its receptor, urokinase plasminogen activator receptor (uPAR) and generation of cell surface localised plasmin (which is protected from inhibition by α2-antiplasmin) instigates multiple extracellular and intracellular (signaling) effects resulting in tissue remodeling and cellular proliferation, cell survival as well as altered cellular adhesion and migration. In cancer, uPAS components including uPA, uPAR and plasminogen activator inhibitor-1 (PAI-1) are upregulated in an uncontrolled fashion and contribute to inappropriate cell signaling and proteolysis. Upregulators of the plasminogen activation system include, but are not limited to, the Epidermal Growth Factor (EGF), Hepatocyte Growth Factor (HGF), Prostaglandin-E2 (PGE-2) and Tumour Growth Factor-beta (TGF-β). See text for details. Created with BioRender.com (accessed on 17 June 2021).

**Figure 2 cancers-13-04097-f002:**
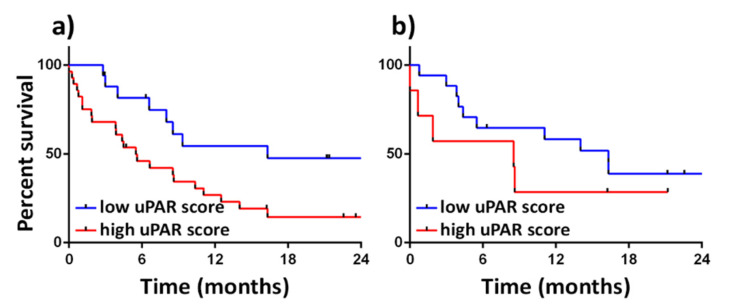
Kaplan-Meier curve showing association of GOC tumour uPAR score and overall survival (OS). Low uPAR (0–1) and high uPAR expression (>2) on tumour cells assessed by immunohistochemistry (IHC) at (**a**) tumour invasion front (n = 43; *p* = 0.02) and (**b**) tumour core (n = 24; *p* = 0.2).

**Figure 3 cancers-13-04097-f003:**
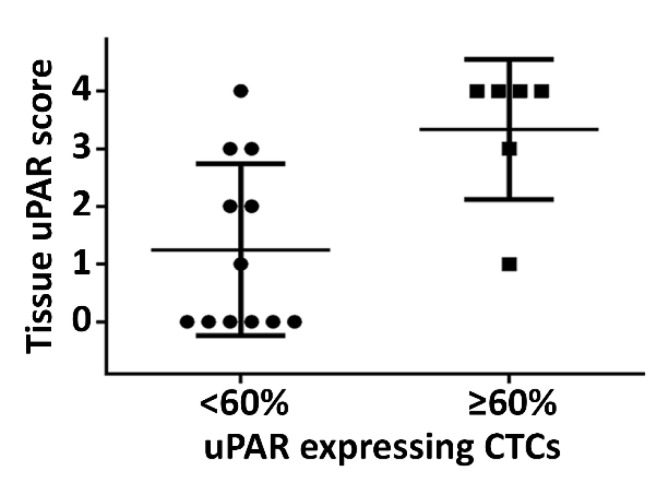
Tissue expression of uPAR is increased in patients where more than 60% of their epithelial cell adhesion molecule (EpCAM) selected circulating tumour cells (CTCs) co-expressed uPAR. Mean score in the lower group 1.3, higher group 3.3 (n = 18, *p* = 0.0008).

**Figure 4 cancers-13-04097-f004:**
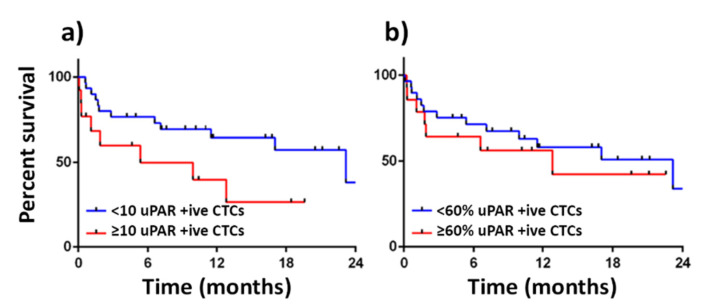
Kaplan Meier curves showing OS demonstrating (n = 43): (**a**) a trend of improved survival seen in patients with less than 10 EpCAM+/uPAR+ CTCs (*p* = 0.06); (**b**) no survival benefit seen where more than 60% of EpCAM+ CTCs also co-expressed uPAR (*p* = 0.5).

**Table 1 cancers-13-04097-t001:** Molecules and pathways linked to regulation of the urokinase plasminogen activation system (uPAS) in gastroesophageal adenocarcinoma cell lines.

Molecule, Pathway	Derived Cell Line ^1^	Effect on uPAS	Reference
**Upregulators**			
Epidermal Growth Factor (EGF) Extracellular cell signalling transducer	SGC-7901 BGC-823	Exogenous EGF increased uPAR mRNA Blockade of ERK1/2 reduced uPAR mRNA expression	Wang, P., et al., 2017 [72]
		Reduced uPA mRNA seen in ERK blockade	Wang, J., et al., 2016 [68]
Osteopontin Linked to PI3K/NFkB/IKK pathways	SGC-7901 BGC-823	siRNA against Osteopontin resulted in reduced uPA mRNA levels. Xenograft model showing reduced tumour growth	Gong, M., et al., 2008 [69]
Hepatocyte Growth Factor (HGF)/cMET pathway Extracellular cell signalling transducer	NUGC-3 MKN-28	Exogenous HGF exposure increased uPA and uPAR protein levels. Blockade of uPAR with antibody or siRNA resulted in reduced wound invasion, which could not be overcome with exogenous HGF.	Kyung Hee, L., et al., 2006 [73]
		HGF signal transduction occurs via JunB/survivin pathway. Survivin inhibition resulted in reduced uPA protein expression and reduced cell invasion.	Kyung Hee, L., et al., 2011 [78]
		MEK inhibition resulted in reduced uPA protein levels, whilst PI3K inhibition showed no change in uPA level. Suggesting uPA activation by HGF via ERK pathway.	Lee, K., et al., 2014 [70]
		HGF signal transduction via PKC/PKD pathway can release HDAC5; HDAC5 increased uPA and MMP-9 activity. Blockade of HDAC5 (even in presence of exogenous HGF) resulted in reduced uPA protein levels. HDAC5-inhibited cells showed reduced cell invasion.	Lee, K., et al., 2010 [66]
COX-PGE2 pathway	AGS	Exogenous prostaglandin E2 resulted in increased levels of uPA and uPAR (protein and mRNA).	Lian, S., et al., 2017 [74]
		Nicotine exposure increased PGE2 resulting in increased uPA and uPAR protein expression	Shin, V., et al., 2005 [75]
Laminin receptor (67LR)	SGC-7901 MKN-45	Downregulation of 67LR resulted in reduced cell line uPA protein expression.	Liu, L., et al., 2010 [79]
TGF-β pathway	SNU-216	Exogenous macrophage inhibitory cytokine 1 (MIC-1; a member of the TGF-β superfamily) resulted in increased uPA and uPAR (mRNA and protein); PAI-1 (mRNA) unaltered.	Lee, D., et al., 2003 [76]
		Interferon gamma inhibition resulted in TGF-B downregulation via smad 2/3 pathway with downregulation of uPA protein expression.	Kuga, H., et al., 2003 [67]
	OE33 FLOW	Increased PAI-1 mRNA levels on exposure via downstream activation of PI3K, ERK and JNK pathways on TGF-β activation.	Onwuegbusi, B., et al., 2007 [71]
**Downregulators**			
p75NTR, NF-κB signalling pathway	SGC7901 MKN45	Upregulation of p75NTR protein caused reduced protein levels of uPA.	Jin, H., et al., 2005 [80]
Tspan9, ERK1/2 pathway	SGC7901	Reduced protein levels of uPA through ERK1/2 blockade.	Li, P. et al., 2016 [77]

^1^ Gastric cancer cell lines: SGC7901 (metastatic, human papilloma virus+), BGC-823 (metastatic, human papilloma virus+), NUGC-3 (metastatic, microsatellite instable, TP53 mutation), MKN-28 (metastatic, microsatellite stable, TP53 mutation), AGS (primary, HPV negative, microsatellite stable) and SNU-216 (metastatic, TP53 mutation). Oesophageal cell lines: OE33 (primary cancer, TP53 mutation) and FLOW (primary, TP53 mutation). Key: uPAS = urokinase plasminogen activation system; uPAR = urokinase plasminogen activator receptor; uPA = urokinase plasminogen activator; PAI-1 = plasminogen activator inhibitor-1; ERK = extracellular signal-regulated kinase; PI3K = phosphoinositide 3-kinase; NFkB = nuclear factor kB; IKK = inhibitor of NFkB kinase; siRNA = small interfering RNA; cMET = mesenchymal epithelial transition factor; MEK = mitogen-activated protein kinase kinase; PKC = protein kinase C; PKD = protein kinase D; HDAC5 = histone deacetylase 5; MMP = matrix metalloproteinase; COX = cyclooxygenase; TGF-β = transforming growth factor-beta; MIC = macrophage inhibitory cytokine; JNK = Jun-N-terminal kinases.

**Table 2 cancers-13-04097-t002:** Key references demonstrating the association of overexpression of each uPAS protein with clinicopathological features.

Clinicopathological Feature	uPA	uPAR	PAI-1
	Key References		
T stage	[22,23,24]	[21,23,26]	[21,23,24,25,26]
Lymph nodes	[22,23,24]	[21,23,26]	[21,23,24,25,26]
Distant metastasis	[22,23,24]		[24]
Vascular invasion	[22,23,24]	[21,23,26]	[21,23,25,26]
Lymphatic invasion	[22,23,24]	[21,23,26]	[21,23,25,26]
Peritoneal disease ^1^	[22,23]		
Serosal involvement ^1^	[22,23]		
Depth of invasion		[21,23,26]	[21,23,24,25,26]
Histological grade		[21,23,26]	[24]

Empty boxes demonstrate no reported evidence found. ^1^ Gastric cancer only. Key: T stage = tumour invasion stage.

**Table 3 cancers-13-04097-t003:** Urokinase plasminogen activation system association with relapse-free- and overall survival (combined immunohistochemistry, in situ hybridisation and enzyme-linked immunosorbent assay data).

	Recurrence-Free Survival HR (95% CI)	Overall Survival HR (95% CI)
uPA	1.90 (1.16–3.11, *p* = 0.01) 3 studies, 467 patients	2.21 (1.74–2.80, *p* < 0.0001) 12 studies, 1094 patients
uPAR	2.69 (NR, *p* = 0.03) 1 study, 203 patients	2.19 (1.80–2.66, *p* < 0.0001) 11 studies, 1036 patients
PAI-1	1.96 (1.07–3.58, *p* = 0.03) 3 studies, 467 patients	1.84 (1.28–2.64, *p* < 0.0001) 10 studies, 839 patients
PAI-2	NR no studies	0.97 (0.48–1.94, *p* = 0.92) 2 studies, 145 patients

CI = Confidence intervals, HR = Hazard Ratio, NR = not reported. Brungs et al. [17].

**Table 4 cancers-13-04097-t004:** Urokinase plasminogen activation system expression in the tumour microenvironment of other solid tumours.

Tissue	Cell Type	uPA	uPAR	PAI-1	PAI-2
Breast, ductal	Macrophages	+ [98]	+ [99,100]	+ [101]	
	Fibroblasts	+ [98,102]	Weak [102]	Weak [102]	
Colon	Macrophages		+ [103]		
	Fibroblasts	+ [104]	+ [103]		
Lung	Macrophages	+ [105]		+ [105]	+ [105]
	Fibroblasts				

Key references noted in each positive box. Boxes unfilled demonstrate no available evidence. Abbreviations: uPA = urokinase plasminogen activator; uPAR = urokinase plasminogen activator receptor; PAI-1 = plasminogen activator inhibitor 1; PAI-2 = plasminogen activator inhibitor 2; + = medium to strong positivity.

## Data Availability

The data presented in this study are available on request from the corresponding author.

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
