# Peer review of "Experimental and Clinical Evidence Supports the Use of Urokinase Plasminogen Activation System Components as Clinically Relevant Biomarkers in Gastroesophageal Adenocarcinoma"

_cancers, 2021, doi:10.3390/cancers13164097_

Round 1

Reviewer 1 Report

This is an excellent review.

Please give some details about the suPAR determination and the available methods IVD and RUO.

Author Response

We thank Reviewer 1 for their review. We have now provided details on suPAR determination and available methods as requested in section 5.1, paragraph 2 of our revised manuscript. We have removed a prior reference of time-resolved fluoroimmunoassay in section 2, paragraph 3. We agree the addition of these methods enhance the understanding and context of this manuscript.

Reviewer 2 Report

The objective of this review article is to summarize the key experimental and clinical evidence on the use of expression patterns of urokinase plasminogen activator system (uPAS) as clinical biomarker in gastric and esophageal cancers (GOCs). The authors have summarized findings from various studies implicating the uPAS system overexpression and their association with poor prognosis in various cancers including GOCs. The authors have also discussed the role of stromal environment, circulating tumor cells, key molecules, pathways, uPAS proteins overexpression in relevance with clinicopathological features of cancer, recurrence free survival (RFS), and overall survival (OS). It is very interesting article but the manuscript could not be considered for publication in the current form for following reasons.

Major concerns:

1) Plasminogen activator inhibitor-1 is supposed to inhibit the activity of uPA–uPAR complex. However it is reported that overexpression of PAI-1 is associated with poor prognosis in cancer. The authors have cited PAI-1 role in various cancer promoting activities but did not comment on rational for this contradicting observation. Could authors please briefly include the details on this contradicting role of PAI-1 in cancer.

2) Could authors also please comment on factors/mechanism that negate the effect of PAI-1 on regulation of uPA-UPAR activities and authors perspective on enhanced uPAS system activity even in the presence of increased PAI-1 in cancer? 

3) Could authors please comment whether there are any studies investigating the therapeutic efficacy of blocking uPA-uPAR interaction in GOCs or other cancers. If so could please include the details?

4) The authors please include a sub-section and discuss about on-going therapeutic efforts on targeting uPAS pathway in GOCs or in the context of other cancers.

Round 2

Reviewer 2 Report

The authors have addressed the concerns and the manuscript could be considered for publication.